# Internet of things–Enabled technologies as an intervention for childhood obesity: A systematic review

Ching Lam[1], Madison Milne-Ives[2], Richard Harrington[3], Anant Jani[4], Michelle Helena van Velthoven[3], Tracey Harding[5], Edward Meinert[2,5,6,7] *

1 Department for Biomedical Engineering, University of Oxford, United Kingdom, 2 Centre for Health Technology, University of Plymouth, United Kingdom, 3 Nuffield Department of Primary Health Care Services, University of Oxford, United Kingdom, 4 Oxford Martin School, University of Oxford, United Kingdom, 5 School of Nursing and Midwifery, University of Plymouth, United Kingdom, 6 Department of Primary Care and Public Health, School of Public Health, Imperial College London, 7 Harvard T.H. Chan School of Public Health, Harvard University, United States of America

* edward.meinert@plymouth.ac.uk

**Data Availability Statement:** All data used was extracted from previously published work and cited accordingly.

## Abstract

Childhood obesity is one of the most serious public health challenges of the 21st century, with consequences lasting into adulthood. Internet of Things (IoT)-enabled devices have been studied and deployed for monitoring and tracking diet and physical activity of children and adolescents as well as a means of providing remote, ongoing support to children and their families. This review aimed to identify and understand current advances in the feasibility, system designs, and effectiveness of IoT-enabled devices to support weight management in children. We searched Medline, PubMed, Web of Science, Scopus, ProQuest Central and the IEEE Xplore Digital Library for studies published after 2010 using a combination of keywords and subject headings related to health activity tracking, weight management, youth and Internet of Things. The screening process and risk of bias assessment were conducted in accordance with a previously published protocol. Quantitative analysis was conducted for IoT-architecture related findings and qualitative analysis was conducted for effectiveness-related measures. Twenty-three full studies are included in this systematic review. The most used devices were smartphone/mobile apps (78.3%) and physical activity data (65.2%) from accelerometers (56.5%) were the most commonly tracked data. Only one study embarked on machine learning and deep learning methods in the service layer. Adherence to IoT-based approaches was low but game-based IoT solutions have shown better effectiveness and could play a pivotal role in childhood obesity interventions. Researcher-reported effectiveness measures vary greatly amongst studies, highlighting the importance for improved development and use of standardised digital health evaluation frameworks.

**Funding:** This manuscript was supported by the Sir David Cooksey Fellowship in Healthcare Translation. The views expressed in the paper belong to the authors and are not necessarily those of the funding body or the authors' University affiliations. The funding body was not involved in the study design, data collection or analysis, or the writing and decision to submit the article for publication.

**Competing interests:** The authors have declared that no competing interests exist.

## Author summary

Obesity is a serious public health concern affecting a growing number of children world-wide and can have long lasting effects on their physical and mental well-being. Health and fitness apps have become an increasingly common means for people to manage their weight and new technologies that can connect to the Internet–such as wearable sensors–are becoming increasingly available. We have conducted a systematic review of studies that described internet-enabled interventions for childhood obesity. In our results and discussion, we provide details of the types of devices, the way they collect, transfer, and analyse data, their aims, and their reported impact. In addition to summarizing the current state of these internet-enabled weight management devices, we discuss areas for future research to improve and better evaluate these devices.

## Introduction

In 2016, more than 1.9 billion adults, 39% of the world's adult population, were overweight or obese [1]. This is leading to increased risks of physical and mental health non-communicable diseases (NCDs) such as type 2 diabetes, hypertension, cardiovascular disease, musculoskeletal conditions, depression and certain types of cancers [2]. Overweight and obese children are five times more likely to be overweight and obese in adulthood [3]. Childhood obesity has become one of the most serious global public health challenges of the 21st century affecting children in both developing and developed countries. In addition to the downstream health effects, evidence is also growing that childhood obesity is associated with an increased risk of developing obesity-related NCDs before adulthood. As well as adverse health outcomes and burdens on health systems, childhood obesity can also result in burdens on the individuals and families, adversely affecting quality of life, mental well-being, and education [4,5].

   Clinic-based interventions seem to have favourable impacts in promoting dietary and life-style changes through behavioural therapies and counselling [2], but face-to-face counselling can be costly and may not always be possible for children in rural areas [3]. A potential solution could be the use of network-connected health Internet of Things (IoT) technologies such as smartphones, wearables, sensors, and linked mobile applications (apps) [6]. Wearables and smartphone apps have been used for tracking physical activity and diet and have shown promise in delivering high-quality care at lower costs [7]. Recent improvement of wearable technologies such as wristwatches with activity sensors has seen them evolve from providing measures of steps, distance, calories, and sleep to measures of activity minutes, heart rate and goal- and target-oriented designs [8]. This has allowed more accurate tracking and provides greater insight into the type and form of physical activity undertaken by the user, such as exercise intensity and metabolic rate [9,10]. Combined with the use of wearables, smartphone apps have shown to be useful in increasing physical activity over the short-term but user engagement tends to decline over a longer period of time [11]. To effectively utilise these technologies for weight management and behaviour change for childhood obesity, network-connected health devices and platforms have been designed to be more appealing, gamified and provide education [12]. However, there have not been any published studies focused on the system designs of these technologies and how they have been evaluated.

### Objectives

This systematic review aims to synthesize the evidence on the use of IoT-enabled technologies as an intervention for childhood obesity. The objectives of this review include to:

- provide an overview of IoT-enabled solutions;

- understand the types of technologies through which data are collected from and the types of data collected;

- identify the networks used for data transfer;

- identify the data analyses processes;

- report how researchers evaluate the feasibility and effectiveness of relevant technologies.

## Results

### Study selection

After the removal of duplicates, 2474 studies were identified from the original database search. Two additional records were identified through reading the references of articles. In total, 23 studies met the eligibility criteria for inclusion in this review. Where the data could be represented numerically, meta-analyses were employed; where the data were heterogeneous, qualitative assessment was conducted. Supporting information includes the Preferred Reporting Items for Systematic Reviews and Meta-Analyses (PRISMA) flowchart for study selection (S1 Fig), the PRISMA checklist (S1 PRISMA Checklist), and the list of included studies (S1 Table).

### Study characteristics

The 23 studies included in this review were published in English between 2012 to 2019 [13–35]. As the objective of this study was to provide an overview of IoT-technologies used as an intervention for childhood obesity, the study types were not restricted to randomised controlled trials. The study was classified as the specified trial type. For instance, O'Malley (2014) described a randomised controlled non-inferiority trial protocol. The study types of the included studies are reported in Table 1. Ridgers et al. conducted two studies using the same IoT platform published as two papers; both of the studies met the inclusion criteria and were included [28,29]. However, to avoid double counting research by the same group on the same IoT platform in meta-analyses, only the latest study was included.

Sample sizes ranged from 10 participants [26] to 988 participants [17], with a total of 3700 participants across the 19 included studies. Four studies involved no participants. The length

**Table 1. Included study types.**

| Trial type | # of studies | Included studies |
|---|---|---|
| Randomised Controlled trial | 3 | Direito 2015 [19], Mendoza 2017 [25], Ridgers 2017 [28] |
| Randomised Controlled non-inferiority trial | 1 | O'Malley 2014 [26] |
| Non-randomised controlled trial | 3 | Yang 2017 [33], Tripicchio 2017 [34], De Cock 2016 [17] |
| Cohort study | 1 | Turel 2016 [32] |
| Mixed methods study | 1 | Taki 2019 [31] |
| Intervention study | 3 | Garde 2015 [20], Bi 2017 [15], Lu 2013 [23] |
| Pilot study | 5 | Caon 2018 [16], Delopoulos 2019 [18], Vazquez-Briseno 2012 [35], Phan 2018 [27], Maramis 2014 [24] |
| Evaluation, usability and feasibility studies | 3 | Lindberg 2016 [21], Ridgers 2018 [29], Svensson 2015 [30] |
| Not applicable (e.g. system design) | 3 | Alahmadi 2013 [13], Lopez 2017 [22], Alloghani 2016 [14] |

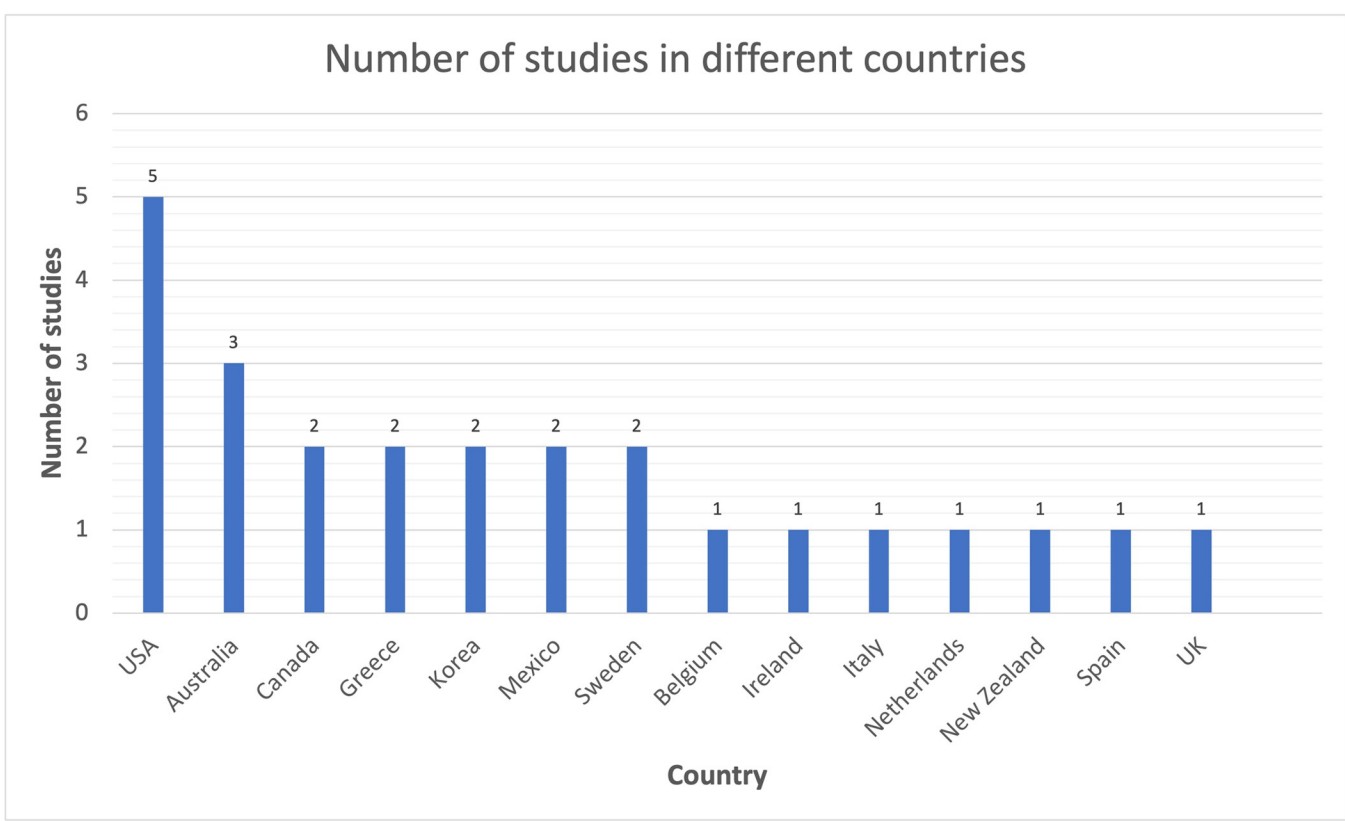

**Fig 1. Studies conducted in various countries.**

of study ranged from several days over a period of one week [21] to nine months [31]. 21 studies took place within a single country, whilst two studies were conducted over multiple countries [16,18]. The number of included studies conducted in each country are listed in Fig 1, with five of the included studies conducted in United States. The Critical Appraisal Skills Programme (CASP) quality appraisal checklists [36] were used for randomised controlled trials, non-randomised controlled trials, cohort studies and qualitative studies and can be found in S2 Table.

## A sensory layer: devices or 'things'

The sensory layer are devices or 'things' that contain sensors which can be used to collect data [37]. This section discusses the device types and the data collected in the included studies.

## Device types

Five device types were identified in the included studies: smartwatch/activity tracker bands, smart garments, smartphone/mobile apps, Near Field Communication (NFC) tags and computer/web services. Some studies employ multiple modalities, for example NFC tags or activity trackers; Fig 2 provides a summary of the number of studies employing each type of device.

Eighteen of the included studies used smartphone embedded sensors or mobile apps as the main intervention. It was noted that some participants did not own a smartphone and had to be provided devices for the study. For instance, in the pilot test conducted in Mexico by

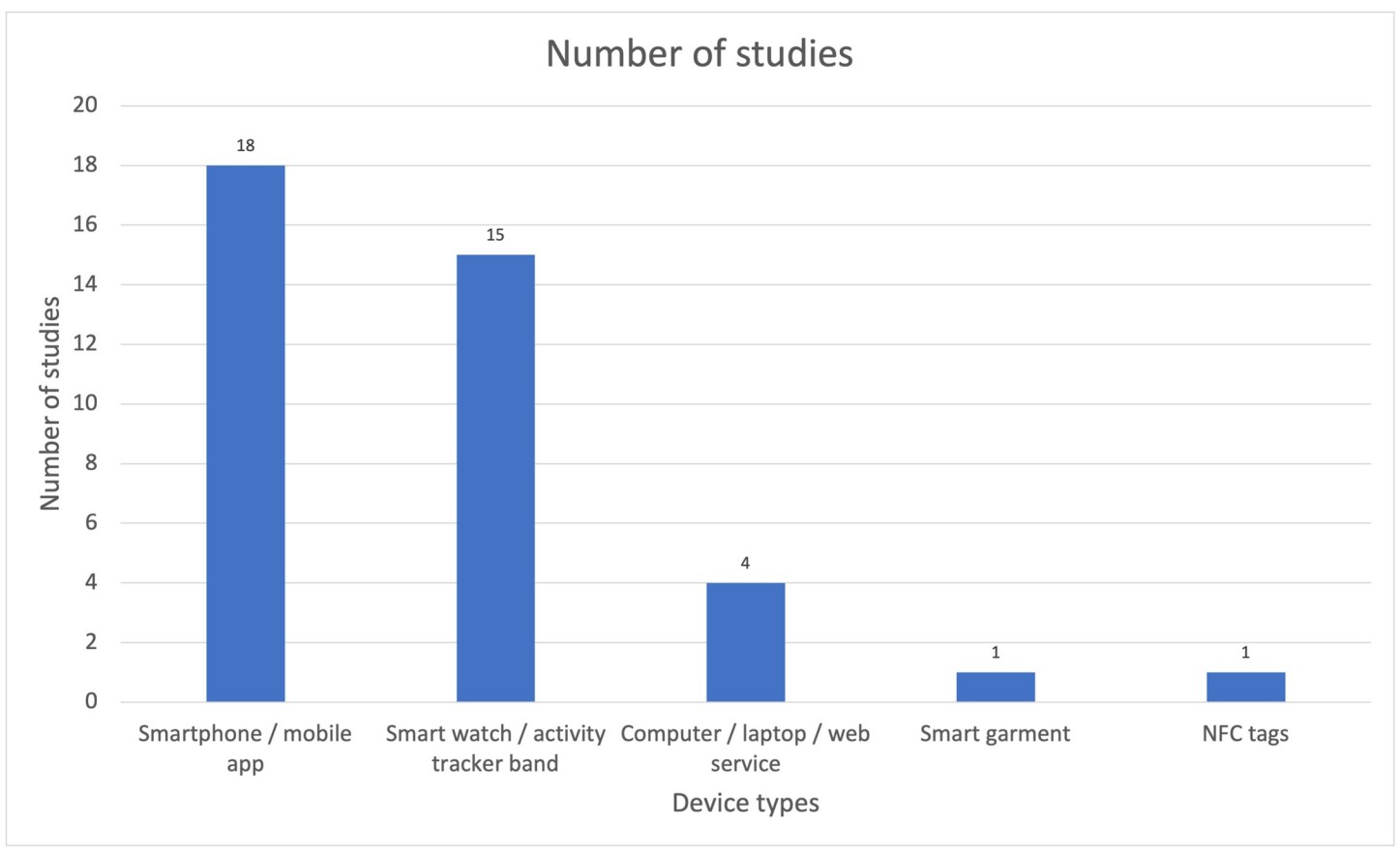

**Fig 2. Device types in included studies**\*. \* The total numbers are higher than the study totals in the review because of overlaps of device type use in studies.

Vazquez-Briseno et al. [35], 2 out of 15 children in the test group had never used or owned a mobile phone before.

Smart watches and activity tracker bands worn on arms, wrists or ankles were the most popular wearables used in the included studies [13,15–25,27,29–30,32,33] (Fig 2). Out of the 8 studies that used commercially available devices, four used a Fitbit (Fitbit, Inc., United States) [25,28,29,32], one used Sensewear Armband (BodyMedia, United States) [30], one used Microsoft Band (Microsoft Corporation, United States) [21], one used Walkie + D Coffee (Green Cross Healthcare Inc., South Korea) and one used Tractivity activity monitor (Kineteks Corporation, United States). The other ten studies used proprietary physical tracking wearables developed for the study. Only one study employed the use of a smart garment which was worn by the user during physical activity [16].

NFC tags were used in one of the studies to enhance a mobile game into an exergame [21], which combines video games with physical exercises [38]. The last type of device in the included studies were computers/web services where users were required to self-report data onto a web portal.

## Sensor type and data collected

Sensors embedded in mobile phones, devices and clothing are used to collect data from the users. The type of sensors that are most frequently used in these devices can be found in Fig 3. The most popular type of sensors were accelerometers (13 out of 23) and Global Positioning

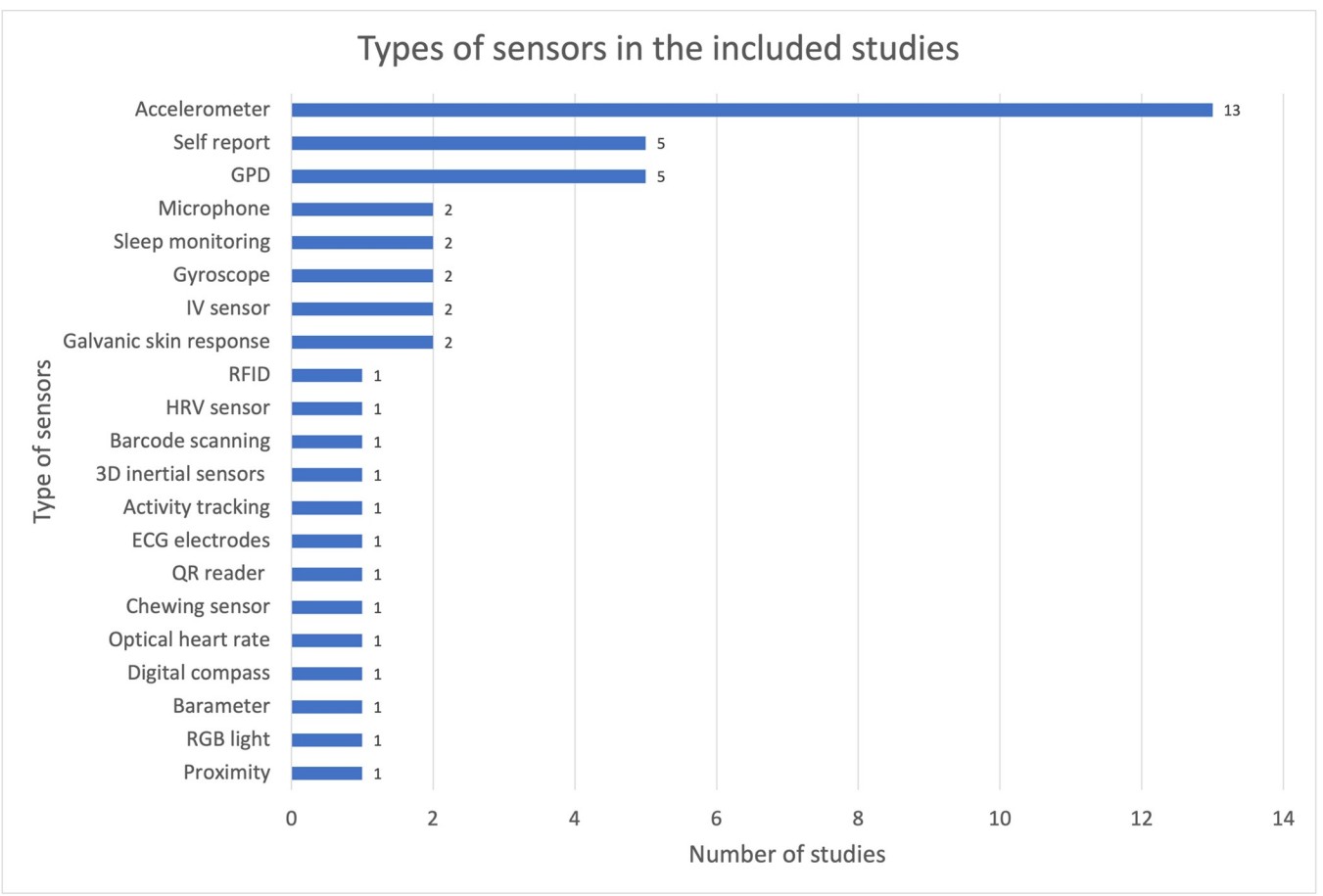

**Fig 3. Sensors in included studies**\*\*.

System (GPS; 5 out of 23), which are critical for physical activity tracking. These sensors are embedded in most smartphones and are used by various mobile apps to collect physical activity data (with the limitation that not all activity is registered). Five of the included studies rely on self-reporting of users to collect data. As shown in Fig 4, physical activity, sleeping patterns and dietary patterns were the most monitored data.

## B Network layer: data transfer

The network layer of the IoT architecture serves as an inter-operation layer for devices to communicate over the internet [39]. For this review, data transfer methods used in the included studies were extracted and are summarised in Fig 5. Data transfer through mobile applications, wireless connection to the internet (via Bluetooth and other unspecified wireless methods) were the most popular options for transfer of data in the included studies.

Only one of the studies used radio-frequency identification (RFID) technology. This method was used for the transfer of TV viewing data, which was collected by detecting nearby television usage from a smart wearable device to a data processing board for further data processing [13].

## C Service layer: data analysis methods

The service layer in IoT architecture handles data processing and analytics to prepare the data for further applications [40]. A summary of the data processing and analysis methods is shown

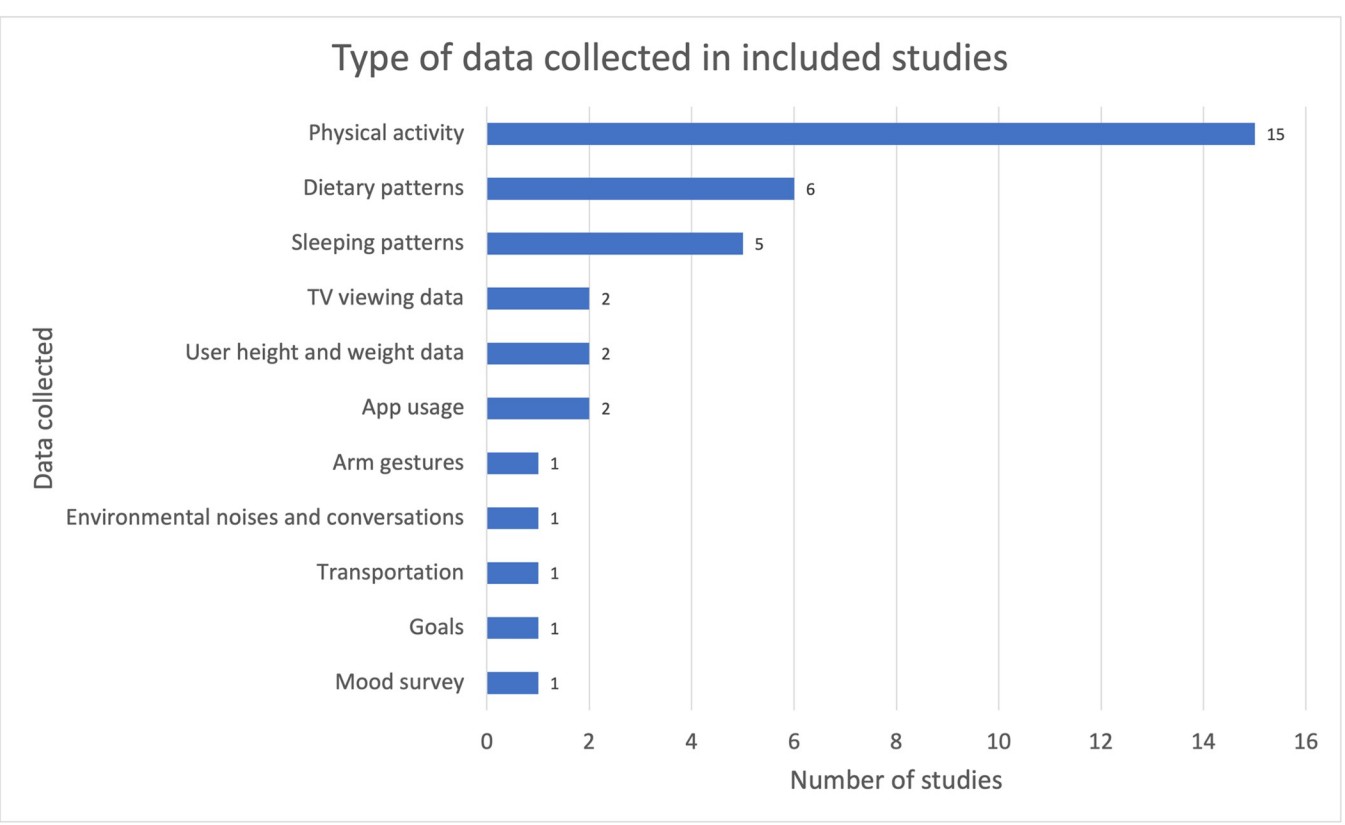

**Fig 4. Type of data collected in included studies**[**]. [**] The total numbers are higher than the study totals in the review because of overlaps of device type and data collection use in studies.

in Fig 6. Over 50% of the studies used simple statistical techniques such as linear regression, T-tests and chi-squared tests as the main method of data analysis, using statistical packages such as Stata version 13SE [17] and SPSS [26,30,33]. Only one study implemented advanced techniques, performing behavioural analyses using signal processing and machine/deep learning [18].

## D Application layer: stakeholders and value delivered to users

The application layer of IoT architecture is where analysed data is applied to meet user needs [40]. Fig 7 shows stakeholders considered in the included studies and the use cases of IoT-enabled technologies are shown in Table 2. 13 out of 23 of the included studies (56.5%) were aimed at monitoring or improving physical activity levels and 8 out of 23 of the included studies (34.8%) were used for monitoring or improving dietary habits.

As Fig 7 demonstrates, although the primary end users of the IoT-enabled technologies were children and youth, these technologies were also used in by adults in family, school, and clinical environments to extend their scope beyond personal monitoring and feedback. For family and caregivers, IoT-enabled technologies allow monitoring of family mealtimes, with an aim to encourage positive dietary habits, increase physical activity levels, and provide associated health education for children. In schools, teachers can be given access to the platform to encourage positive behavior [16].

Together, hospitals, weight management clinics and health professionals were another important stakeholder that was considered in 9 out of 23 of the studies. IoT-enabled

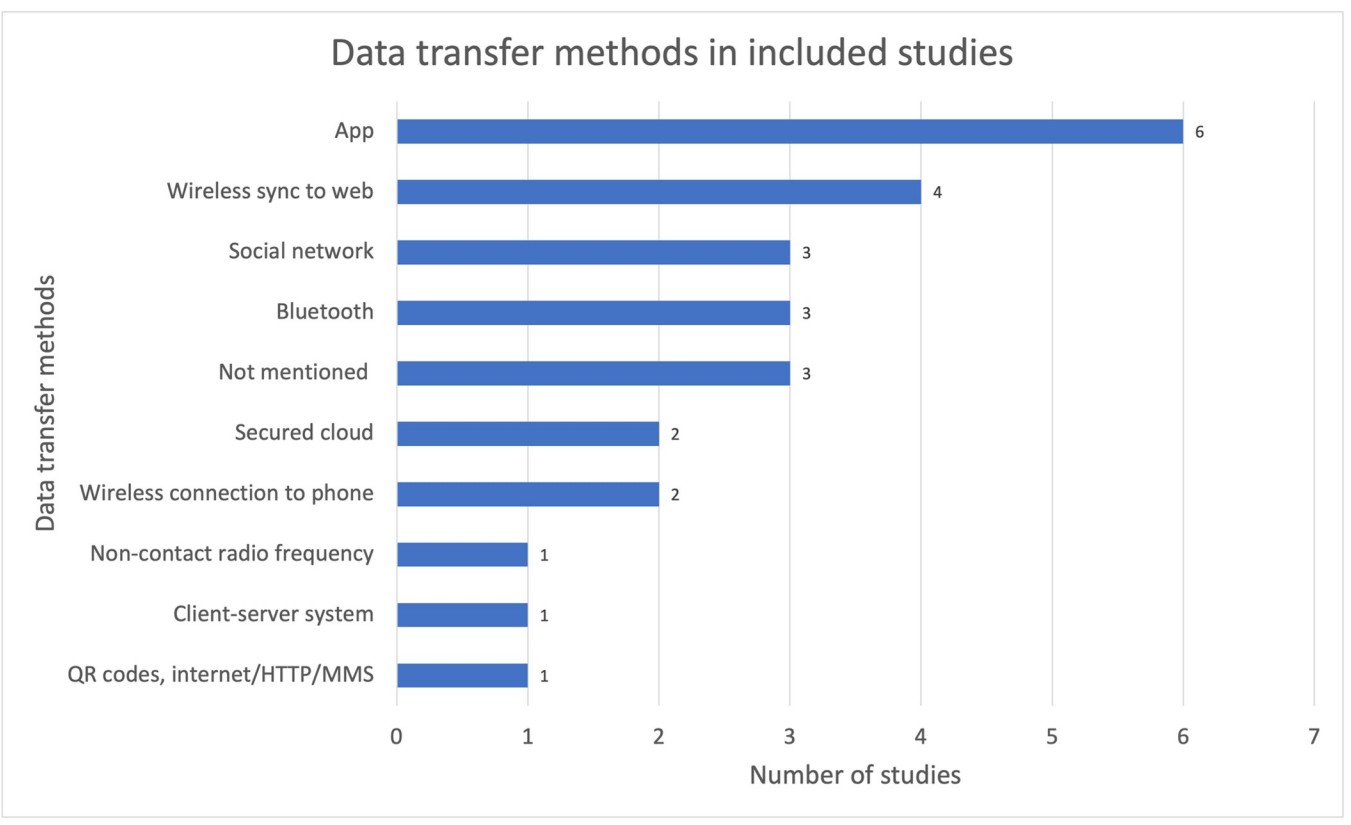

**Fig 5. Data transfer methods in included studies.**

technologies were used by health professionals to set up nutrition and exercise targets for youth, estimate risk and goal achievement, follow up on patient progress, monitor patients remotely and deliver secure and effective remote care. Only one of the studies cited obesity and eating disorder prevention as one of the objectives of hospitals [24].

For researchers, four studies cited the use of IoT for behaviour monitoring. And from a public health perspective, data collected from these user-driven technologies can help policy makers understand diet and physical activity level demographics to drive evidence-based policy changes. S3 Table shows a matrix of the use cases for the various stakeholders reported in the included studies.

### E Behavioural change theories

Various behaviour change theories were used in the included studies and are summarized in Table 3. Monitoring (91%) and feedback (45%) were the most commonly used behavioural change techniques for the included studies, followed by goal setting and action planning (31.8%) and positive reinforcement (e.g. rewards and challenges; 31.8%). Monitoring refers to the use of self-reports or trackers to monitor user habits such as screen time, physical activity and diet. Monitoring can be done by the child themselves or by other stakeholders, such as other family members, teachers, researchers,

and health professionals. Feedback in the context of the included studies refers to technology generated feedback such as weekly progress reports.

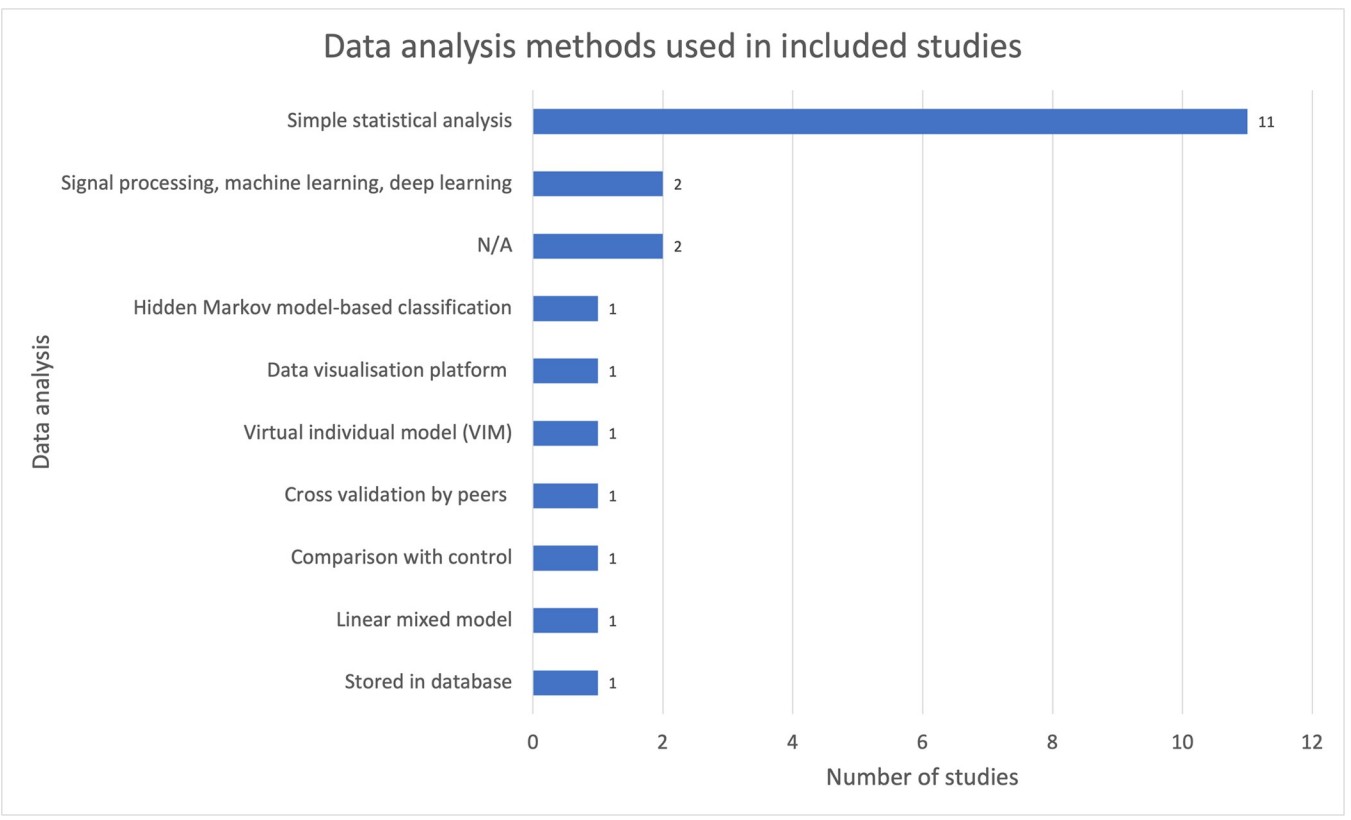

**Fig 6. Data analysis methods used in included studies.**

### F Researcher-reported effectiveness measures and outcomes

As the type of studies included in this review were very heterogenous, the researcher-reported effectiveness measures and outcomes varied greatly between studies (Table 4). Four studies described system designs and therefore are not listed in the table [13,14,18,22]. Despite the heterogeneity of effectiveness and outcome measures, a few subthemes could be identified: accuracy of technology (n = 2 [15,30]), usability and engagement (n = 7 [14,16,17,21,27,29,31]), qualitative perception and acceptance of users towards technology (n = 9 [14,19–21,25,26,29,31,33,34]), behaviour change metrics (e.g. improved physical activity level, n = 8 [17,18,20,25–27,29,33])) and physical improvement (BMI change, n = 7 [19,21,23,26,32–34]). Only Direito et al. [19] reported their results based on the Consolidated Standards of Reporting Trials of Electronic and Mobile HEalth Applications and onLine TeleHealth (CONSORT-eHealth) checklist [41] aimed at standardising eHealth and mHealth trial reporting.

### G Ethics and regulations

A summary of studies that mentioned ethical approval and participant consent can be found in S4 Table. For studies detailing system designs, institutional review is not necessary. However, different countries and universities may have slightly different requirements for ethical review and participant consent.

With regard to regulations, only one of the included studies mentioned compliance with EU Directive 95/46/EC and General Data Protection Regulation (GDPR) [18], and none of the studies included mention the Health Insurance Portability and Accountability Act (HIPAA),

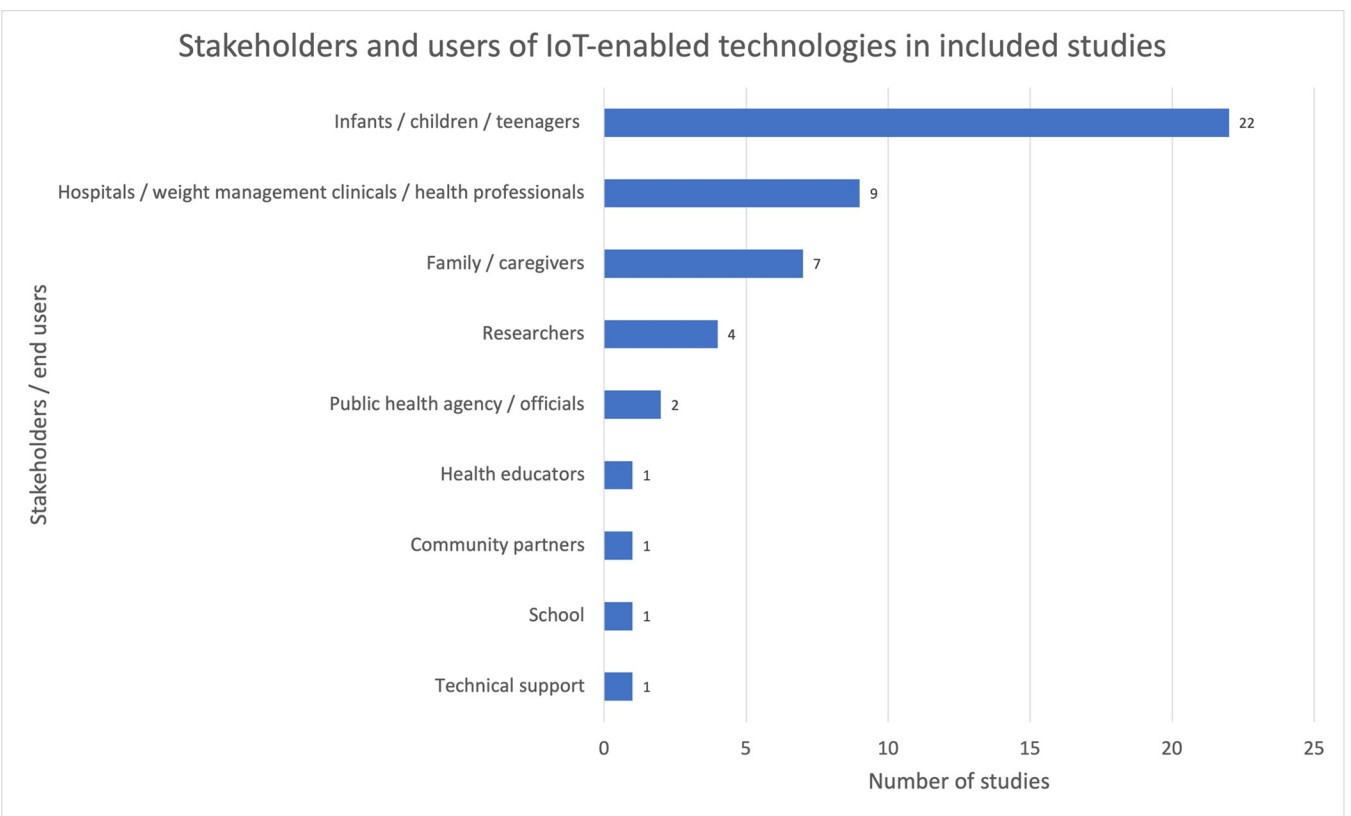

**Fig 7. Stakeholders and users of IoT-enabled technologies in included studies (number in bracket indicates the number of studies which considered the stakeholder).**

which is an important healthcare data protection law in the US. Only two of the included studies mentioned the use of a "secure cloud platform" [16,18] and one mentioned compliance with university data protection regulations [26].

## Discussion

### Principal findings

This meta-analysis identified the current state of art for IoT-enabled weight management interventions for youth. A summary of the IoT architecture of the included studies can be found in Fig 8. The current technologies have different levels of effectiveness consistent with

**Table 2. Use cases for infants/ children/ teenagers.**

| Use case: infants / children / teenagers | Studies |
|---|---|
| Screen time monitoring (television, mobile phones, video games, etc) | [13,15,32,33] |
| Sleep pattern monitoring | [18,32,33] |
| Dietary behaviour monitoring | [16,18,30,33,35] |
| Food advertisement exposure monitoring | [18] |
| Snacking behaviour monitoring | [17] |
| Improve dietary habits | [22,31] |
| Physical activity monitoring | [18,30,33] |
| Improve physical activity level | [19–22,25,27,29] |

**Table 3. Use cases for infants/ children/ teenagers.**

| Theory or technique | [35] | [22] | [19] | [23] | [28] | [30] | [20] | [13] | [26] | [27] | [28] | [18] | [29] | [17] | [32] | [33] | [25] |
|---|---|---|---|---|---|---|---|---|---|---|---|---|---|---|---|---|---|
| Monitoring | * | * | * | * | * | * |  | * | * | * | * | * | * | * | * | * | * |
| Goal setting and action planning |  | * | * |  |  |  |  |  |  | * |  |  | * | * |  |  |  |
| Prompts and cues |  |  |  |  |  |  |  |  |  |  |  |  |  |  |  |  |  |
| Positive reinforcement (e.g. rewards and challenges) |  | * | * |  |  |  |  |  | * |  |  | * | * | * |  |  |  |
| Self determination |  |  |  |  |  | * |  |  |  |  |  |  |  |  |  |  |  |
| Self- regulatory |  |  |  |  | * |  |  | * |  |  |  | * |  |  |  |  |  |
| Feedback |  | * | * |  | * | * |  | * |  | * |  | * |  |  |  |  | * |
| Gamification |  |  |  |  |  |  | * |  | * |  |  |  |  |  |  | * |  |
| Active learning and education |  |  | * |  |  |  |  |  |  |  |  | * |  |  |  |  |  |
| Advanced organisers |  |  | * |  |  |  |  |  |  |  |  |  |  |  |  |  |  |
| Policy change |  |  |  | * |  |  |  |  |  |  |  |  |  |  |  |  |  |
| Socialisation |  |  |  |  |  |  |  | * | * |  | * | * |  |  |  |  |  |
| Parent role modelling |  |  |  |  |  |  |  |  |  |  |  |  |  |  |  |  |  |

what has been previously reported in the literature, as there is a large degree of heterogeneity in the outcome measures/study types analysed in this paper, and so different levels of effectiveness observed may be due to these inconsistencies / flaws in experimental design [42,43].

There was also a large variety in the sensors, data transfer methods, and methods of analysis used in the different IoT technologies. The number of sensors and data transfer methods highlight the possibilities and flexibility that IoT technologies can have; there are many ways of designing them to address specific needs and contexts. Understanding these needs and contexts will be important for developers, to ensure that their technology can be easily adopted and implemented. Few of the IoT technologies used advanced analytic techniques such as machine learning; this is somewhat surprising, as interest and use of artificial intelligence and machine learning in healthcare is growing rapidly. It is possible that IoT technologies incorporating these methods are still in early stages, but future research should examine the potential impact the inclusion of artificial intelligence and machine learning could have on supporting engagement, analysing user data, and personalising interventions.

Multiple studies cited the importance of cheap and accessible ways for monitoring obesity-related behaviours through the use of IoT solutions [13,28,31,32]. For some studies, children did not own smartphones and the smartphones had to be provided to the children for the studies [17]. Whilst mobile apps can be cheap or free, the requirement of a card for app payment has been cited as a barrier for accessibility [19]. Therefore, when designing youth-specific weight management systems, it is important to consider the smartphone ownership demographics in the country and the age-group that the system is designed for.

As shown in Table 4, the engagement rates of interventions involving games are more effective, especially the exergame designed by Lindberg et al. [21]. This highlights the importance of designing engaging interventions for children to keep users from dropping out and improving adherence in real environment where children interact with others face-to-face may be one of the factors leading to the better perception of the technology.

IoT technologies allowed stakeholders such as families and schools to monitor the habits of the users through data continuously generated from the technologies. A better understanding of the obesity-related behaviours of children and young people, through the data generated by IoT platforms, can be invaluable. Various projects (Big O [18], SPLENDID [24] and PEGASO [16]) are ongoing to test the usability and usefulness of these technologies.

**Table 4. Researcher-reported outcomes.**

| Study | Researcher-reported effectiveness measures and outcomes | Reported results |
|---|---|---|
| [15] | Precision (ratio of the number of true-positive windows to total number of windows) | For meal detection: 88.7%<br>Family activity detection: 97.8% |
| | Recall (number of true-positive windows divided by the total number of windows detected as family meals) | For meal detection: 93.3%<br>Family activity detection: 92.8% |
| [16] | Subjective rating of product's usability using the System Usability Scale (SUS) | 64.66/100 |
| | Emotional outcomes of interaction between user and product using Emotional Metric Outcomes (EMO) | 6.5/10 |
| [17] | • Healthy snack ratio<br>• Awareness<br>• Intention to eat healthy<br>• Attitude regarding taste of healthy snacks<br>• Self-efficacy to eat healthy<br>• Habit to eat healthy | Statistically insignificant between intervention (with reward-based app) and control group (usual curriculum) |
| | • Attitude regarding overall health<br>• Knowledge about healthiness of snacks | Slight improvement |
| | Engagement after 4 weeks | 20.5% |
| [19] | Time to complete 1-mile walk/run fitness test | Compared to control:<br>Immersive app: -28.4s Non-immersive: -24.7s |
| | • Perceived enjoyment<br>• Perceived competence, autonomy and relatedness<br>• Self-efficacy | No intervention effects found |
| [20] | Intervention effect in activity | Significant increase -<br>Game group: steps increase 1191 daily, 25 active minutes increase<br>Feedback group: 796 steps increase, 6 more active minutes. |
| | Enjoyment | 16% brilliant, 42% very good, 23% good, 15% not very good, 4% awful |
| [21] | Learning: pedagogical quiz correct answers | Game group: 77%<br>Control group: 65% |
| | Exercise efficiency: heart rates of users | Average game session lasted for 8 minutes.<br>Maximum = 11bpm<br>Minimum = 65.9bpm<br>Average = 83.7bpm |
| | Motivation and engagement: qualitative interviews<br>• RO2 was more exciting than normal computer or mobile games<br>• RO2 was more exciting than physical education classes<br>• After playing RO2, doing sports is more interesting than before | Strongly agree/agree:<br>• 24/32<br>• 27/32<br>• 28/32 |
| [23] | BMI reduction | Average BMI decrease of 0.51 for overweight/obese |
| [25] | Moderate-to-vigorous physical activity (min/day) | No significant difference |
| | Sedentary time (min/day) | Intervention: - 4.5 min<br>Control: + 1 min |
| | Patient-reported experience (helpfulness of intervention for PA awareness, setting personalised goals, motivation and joining virtual community of survivors with common goal) | Majority felt that the intervention was helpful |

(*Continued*)

**Table 4.** (Continued)

| Study | Researcher-reported effectiveness measures and outcomes | Reported results |
|---|---|---|
| [26] | • Anthropometric parameters (weight, height, BMI) <br>• Blood sample analysis <br>• Insulin resistance <br>• Blood pressure <br>• Physical activity <br>• Cardiorespiratory fitness <br>  Muscoskeletal screen <br>• Health-related quality of life <br>• Psychosocial health <br>• Technical effectiveness <br>• Relative user efficiency <br>• User satisfaction | N/A (design) |
| [27] | Fitness tracker utilisation | 69% discontinued use by end of study |
|  | Users felt the tracker was helping them meet lifestyle goals | 69% |
|  | Users felt more motivated to achieve a healthy weight | 66% |
|  | Daily steps on valid device days | Continued usage resulted in higher step count |
|  | Daily minutes of moderate to vigorous physical activity on valid device | Continued usage group: lower initial use, with a further reduction over 3 months |
|  | Daily calories burned on valid device | Continued usage group: higher |
| [29] | Qualitative insights from subjects regarding views on the acceptability and usability of the different features and functions, potential facilitators or barriers to ongoing use, thoughts on impact of Fitbit on overall activity levels | Easy to use for activity tracking, more difficulty with monitoring sleep <br>Reported barriers in usability |
| [18] | Self-reported measures such as: <br>• Number of days physically active <br>• Time spent on sedentary behaviour <br>• Usual mode of transport <br>• Self-efficacy <br>• Social support | N/A (Trial results not published) |
| [30] | Accuracy of assessment of total energy expenditure of users | Mobile phone app underestimated energy expenditure by 29% compared to wearable |
| [31] | Engagement index (EI) | Average = 36.2% |
|  | Participant experience | Broad themes and subthemes identified |
| [32] | Videogame addiction, obesity level | Videogame addiction negatively associated with sleep duration. |
| [33] | Primary outcome measures: behavioural changes (including healthy eating, increased physical activity, and fitness) | N/A (design) |
|  | Secondary outcome measures: <br>• Changes in anthropometric parameters (body weight, height, body mass index z-score, and waist circumference) <br>• BMI percentiles (obesity rate) <br>• Psychological perceptions among participants | N/A (design) |
| [34] | Child BMI z-Score | No significant reduction |
|  | Treatment acceptability | Highly acceptable |
| [35] | Barcode readers acceptance | Low to excellent for different barcode readers |
| [14] | Mutual relationship of system's measurement variables that include trust, security, ease of use and usefulness | Positive correlation between trust and security, ease of use and security, usefulness and ease of use |

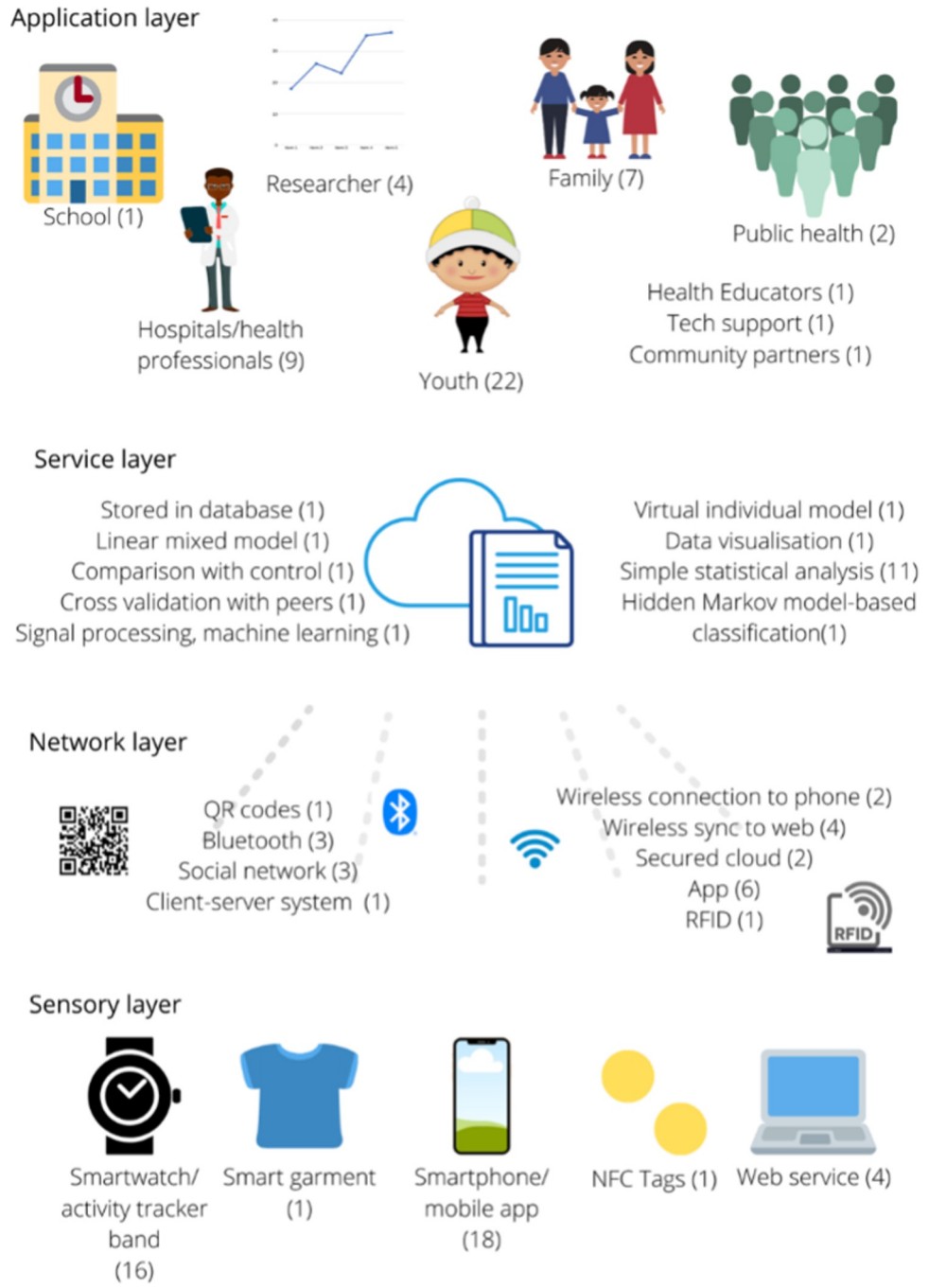

**Fig 8. Summary of IoT architecture of included studies.**

Whilst there are many studies ongoing in this field, the general lack of adherence to a standardised evaluation framework has been noted. Only one study used the CONSORT e-Health standards for evaluation, an extension of the CONSORT checklist for improving reporting of randomized controlled trials of eHealth and mHealth interventions [41]. This highlights the importance of raising researcher awareness to improve evaluation protocol designs, as the use of this checklist provides a standardized means of evaluating the validity of studies of eHealth interventions.

## Strengths and limitations

This study is the first to look at the application of IoT-enabled technologies specifically for weight management in children and adolescents. This landscape study collated the effectiveness and results of previous studies and the results of our systematic review highlight some options that may be useful for future development of IoT-enabled technologies for children. A further strength is the mapping of the overall growth and acceptance of sensors / IoT / platforms / data processing, but also the rapid pace of change in technologies, marketplace and devices. However, the limited number of included studies and the heterogeneity of study types and intervention formats made it difficult to do rigorous cross-study comparisons using the types of data extracted for meta-analysis or to conduct a rigorous investigation of the potential impact of various features, such as gamification.

## Future recommendations

The results of our study suggest that the future design of IoT-enabled weight management strategies for children and families should explore low-cost but engaging game-based technologies. Previous research has demonstrated that gamification can be an effective means of increasing engagement with digital interventions [44]. However, there is a still a lack of empirical evidence on the impact of gamification [45], especially over the long term [46], and interventions that include gamification should examine effectiveness over time as well as any potential unintended effects that could arise from its use. There is insufficient evidence to show good engagement for non-game-based technologies for children.

To reduce the heterogeneity of studies examined, future reviews could narrow their scope to focus on particular types of IoT-enabled technology or limit their search to a particular study type (such as randomized controlled trials). This could provide a more homogenous and methodologically strong set of studies, which could enable more robust conclusions to be drawn. Future studies should also aim to adhere to standardised evaluation methods in order to allow better comparison between studies and should conduct more holistic evaluations of their technologies.

## Conclusions

This is the first systematic review focusing on the application of IoT technologies for weight management of children and youth. 23 studies were included in this systematic review and the IoT framework of these studies has been extensively studied. In the sensory layer, smartphone/mobile apps were the most commonly used devices (78.3%) and physical activity data (65.2%) from accelerometers (56.5%) were the most commonly tracked data. In the network layer, mobile app-based data transfer (26.1%) was the most commonly used method of data transfer. In the service layer, over 50% of the studies employed simple statistical analysis whilst only one study embarked on machine learning and deep learning methods. In the application layer various stakeholders and user-specific use cases were identified. A variety of outcomes were examined in evaluations of the technologies, but tended to focus on engagement, acceptability, and behaviour and physical changes.

Game-based interventions have shown better effectiveness, but a decline of engagement in technology adoption was observed. Future design of IoT-enabled weight management technologies aimed at children should consider the requirement for a low-cost yet engaging game-based intervention and the general need for a more standardized effectiveness evaluation method.

## Methods

This systematic review was completed in accordance with the PRISMA guidelines [47].

A systematic search was conducted on 16th September 2019 and the review was conducted according to a previously published systematic review protocol [48]. Six electronic databases were searched: Medline, PubMed, Web of Science, Scopus, ProQuest Central and the IEEE Xplore Digital Library. The search strategy, eligibility criteria, PICO, and search terms are detailed in the protocol [48].

### Search strategy

Medline, Pubmed, Web of Science, Scopus, ProQuest Central and the IEEE Xplore Digital Library were searched for studies published after 2010 using a combination of keywords and subject headings related to health activity tracking, weight management, diet, children and IoT (Table 5). The search string was structured with the format: (Health activity tracking) AND (Youth) AND (IoT).

### Eligibility criteria

*Population*: Children (under 18 years) who have interacted with IoT-enabled technologies for weight management, physical activity tracking, and encouragement of a healthy lifestyle.

*Intervention*: IoT-enabled technologies as an intervention for childhood obesity, e.g. smartphones, standalone or embedded physical activity tracker, weight tracking, food trackers, sleep trackers, and computer/web services.

*Comparator*: Studies with and without a comparator will be included.

Outcome:

1. types of weight management technologies and products available for children

2. benefits and limitations of each of these technologies.

3. the effectiveness measures used by researchers and the reported effectiveness of these technologies.

*Study*: All studies will be eligible for inclusion if they describe or evaluate an IoT weight management intervention for children. Review papers, study protocols, and studies not available in English will be excluded.

The inclusion and exclusion criteria for the study can be found in Table 6:

### Data management and selection process

All search results were exported into a Mendeley library and duplicates were removed. To avoid risk of bias, two authors (CL and MMI) independently screened the titles and abstracts

**Table 5. Search terms.**

| Theme | Search terms |
|---|---|
| Health activity tracking device | electronic track* OR (electronic activ* AND track*) OR (electronic activ* AND monitor*) OR electronic fitness track* OR fitness track* OR (wearable AND track*) OR wearable OR sens* |
| Weight management | (Weight AND (manag* OR monitor* OR reduc* OR loss OR maint*)) OR ("body mass index" OR BMI) OR diet OR obes* |
| Children | Child* OR teen* OR youth OR paed* OR ped* OR adolescent* OR young* |
| IoT | IoT OR Internet of things OR data analy* OR data collect* OR connected health OR digital health OR mobile health OR mhealth |

**Table 6. Study inclusion and exclusion criteria.**

| Inclusion criteria | Exclusion criteria |
| --- | --- |
| • Publications written in English<br>• Studies published between 2010 and September 16, 2019<br>• Studies that describe a wearable device or mobile app for health activity tracking connected to an internet platform<br>• Studies that describe the data analysis process and how data are connected to the wider internet | • Studies that do not describe any weight management intervention<br>• Studies that describe devices or apps that are not connected to a wider internet-based platform for data analysis<br>• Studies that are not focussed on children<br>• Review papers and study protocols<br>• Not published in English |

identified from the search and accepted or rejected the studies according to the study inclusion and exclusion criteria. Any discrepancies were discussed and resolved. If consensus was not reached, a third reviewer (EM) was consulted. The full articles for the selected studies were downloaded and analysed by CL to confirm eligibility for analysis. CASP checklists were used to appraise the quality of the included studies where applicable [36].

## Data extraction

CL extracted the data from the included studies. Data from eligible publications were extracted into a predesigned form (which can be found in S5 Table) to identify the types of device or app, IoT architecture employed, type of data collected, data transfer methods, data analysis methods, effectiveness as an intervention for childhood obesity and relevant ethics and regulation if mentioned.

## Supporting information

**S1 PRISMA Checklist. PRISMA checklist.**
(DOC)

**S1 Fig. PRISMA flowchart.**
(DOCX)

**S1 Table. Included studies.**
(DOCX)

**S2 Table. CASP Quality Appraisal Checklist.**
(XLSX)

**S3 Table. Use case matrix.**
(XLSX)

**S4 Table. Mention of ethics in included studies.**
(DOCX)

**S5 Table. Data extraction table.**
(XLSX)

## Author Contributions

**Conceptualization:** Ching Lam.

**Formal analysis:** Ching Lam.

**Investigation:** Ching Lam, Madison Milne-Ives.

**Supervision:** Edward Meinert.

**Writing – original draft:** Ching Lam.

**Writing – review & editing:** Ching Lam, Madison Milne-Ives, Richard Harrington, Anant Jani, Michelle Helena van Velthoven, Tracey Harding, Edward Meinert.

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
