## [Decision Letter · Decision Letter 0]

21 Dec 2021

PDIG-D-21-00086

Internet of things-enabled technologies as an intervention for childhood obesity: A Systematic Review

PLOS Digital Health

Dear Dr. Meinert,

Thank you for submitting your manuscript to PLOS Digital Health. After careful consideration, we feel that it has merit but does not fully meet PLOS Digital Health's publication criteria as it currently stands. Therefore, we invite you to submit a revised version of the manuscript that addresses the points raised during the review process.

Reviewers note several notable strengths including that the premise and focus of the paper is relevant and interesting, the use of attention to gamification, and use of appropriate methods. 

The reviewers note weaknesses that should be addressed in a resubmission. Major comments include but are not limited to: 

- Potentially over-restrictive search strategy and inclusion criteria (see comment re: gaming consoles). If not included in a revision, please add to the limitation that gaming consoles were not in the literature and/or included. 

- The conclusion regarding gaming features is based on limited evidence. Additionally, please include the potential pros and cons of gamification, specifically the risks associated with social networking features. If these potential harms have not been evaluated, please comment on that gap. 

- Results tables are difficult to interpret. Formatting for readability will help. Utilization of capitalization in headers and tables should be addressed as well as other typographical and copy-editing issues. 

- Include in the conclusion what the goals of the study were, only change of weight? Change of lifestyle? Change of behavioral aspects?

We look forward to receiving your revised manuscript.

Kind regards,

Shelagh Mulvaney

Academic Editor

PLOS Digital Health

Journal Requirements:

1. Please update the completed 'Competing Interests' statement, including any COIs declared by your co-authors. If you have no competing interests to declare, please state "The authors have declared that no competing interests exist". Otherwise please declare all competing interests beginning with the statement "I have read the journal's policy and the authors of this manuscript have the following competing interests:"

2. Please provide a complete Data Availability Statement in the submission form, ensuring you include all necessary access information or a reason for why you are unable to make your data freely accessible. Note that it is not acceptable for the authors to be the sole named individuals responsible for ensuring data access.

PLOS defines a study's minimal data set as the underlying data used to reach the conclusions drawn in the manuscript and any additional data required to replicate the reported study findings in their entirety. Any potentially identifying patient information must be fully anonymized. 

If your research concerns only data provided within your submission, please write "All data are in the manuscript and/or supporting information files" as your Data Availability Statement.

3. Please provide separate figure files in .tif or .eps format only and remove any figures embedded in your manuscript file. Please ensure that all files are under our size limit of 20MB. 

For more information about how to convert your figure files please see our guidelines: https://journals.plos.org/digitalhealth/s/figures

Once you've converted your files to .tif or .eps, please also make sure that your figures meet our format requirements.

Additional Editor Comments (if provided):

- The manuscript has several strengths including standardized rubrics, inclusion of the use of empirically-supported behavior change methods, and substantial supporting documentation for processes and outcome of the study.

Additional weaknesses noted here should be addressed for a revision: 

- In the abstract, the stated purpose of the review includes evaluation of feasibility and effectiveness but does not include evaluation of ‘system designs’ (data processes, networks) as stated later in the paper. 

- The authors include stakeholders involved in studies, but who were actually the users of the systems? Were stakeholders involved in any aspect of design but not the final intended users? 

- Please explain why 4 studies had no participants for an intervention study.

- A suggestion to improve the readability of the tables: Table V does not provide a clear indication of how many studies used a particular outcome. Better explanation in the text may also help with that table.

- If the authors would like to put forward the recommendation for developers to focus on gamification for engagement, please add additional support/rationale from the research literature and what features have been found to be most helpful.

- The implications of the results do not come through in the Discussion. What are the implications of the data transfer methods or analyses used to provide data summaries to users? Please describe the possible negative impacts of gamification if social networking is included. Please recommend how future literature reviews could aggregate and organize heterogeneous studies to provide more generalizable results.

Reviewers' comments:

Reviewer's Responses to Questions

**Comments to the Author**

1. Does this manuscript meet PLOS Digital Health’s publication criteria? Is the manuscript technically sound, and do the data support the conclusions? The manuscript must describe methodologically and ethically rigorous research with conclusions that are appropriately drawn based on the data presented.

Reviewer #1: Yes

Reviewer #2: Yes

Reviewer #3: Yes

2. Has the statistical analysis been performed appropriately and rigorously?

Reviewer #1: N/A

Reviewer #2: Yes

Reviewer #3: N/A

3. Have the authors made all data underlying the findings in their manuscript fully available (please refer to the Data Availability Statement at the start of the manuscript PDF file)?

Reviewer #1: Yes

Reviewer #2: Yes

Reviewer #3: Yes

4. Is the manuscript presented in an intelligible fashion and written in standard English?

Reviewer #1: Yes

Reviewer #2: Yes

Reviewer #3: No

5. Review Comments to the Author

Reviewer #1: The authors review novel interventions using internet-connected devices to tackle the important problem of childhood obesity. They review studies focusing on a wide variety of devices, including wristbands and smartphone apps, and present the technologies, data analysis processes and effectiveness in an accessible manner. The precise reporting of what was measured and evaluated for various stakeholders in each study in the use case matrix is appreciated - this will enable interested readers to understand the approaches, strengths and weaknesses at a glance.

The authors correctly note that game-based interventions appear to result in better engagement, and there is insufficient evidence to show good engagement for non-game-based technologies in children. However, their recommendation to ‘… concentrate on low-cost but engaging game-based technologies’ fails to mention the fact that potential harms have not been systematically evaluated - devices used to bring these interventions to children can also expose them to other games and applications like social networks, which can have adverse effects on physical and mental health, as well as social relationships. It may well be that these game-based technologies will prove to be beneficial on balance, but their future evaluation should not skip over the consideration of these risks in real world settings. While a detailed discussion of these issues is probably beyond the scope of this work, it is important that readers understand that the potential downsides of these approaches have not been studied in the reviewed papers.

Reviewer #2: This original research article “Internet of things-enabled technologies as an intervention for childhood obesity: A Systematic Review” is a well written systematic review aiming to assess the current evidence regarding utility of network-connected health Internet of Things (IoT) technologies (smartphones, wearables, sensors, apps, etc) in children and adolescents as a tool for managing and preventing childhood obesity. They appropriately comment on the mental and physical health problems that come along with childhood obesity and the downstream effects it has. They comment on clinic-based interventions having favorable impacts, but note that in person counseling may not be feasible for children in rural areas or with financial difficulties. They propose utilizing IoT devices as a tool for childhood obesity and a solution for location and financial barriers to in person diet/lifestyle counseling. Their analysis involves 23 studies from 2010-2019. The results highlight the most used devices (smartphone/mobile apps) and the most commonly tracked data (physical activity data, accelerometer data). Results indicate that adherence to IoT based approaches was low but improved with game-based interfaces. They conclude that future IoT enabled weight management technologies aimed at children should focus on low-cost, game-based intervention (“gamification”).

The premise of this paper is interesting and addresses a very important topic in the health of children and focus on early intervention for childhood obesity and preventive medicine. The methods used are appropriate, though search criteria may have been restrictive and excluded potentially relevant studies (https://pubmed.ncbi.nlm.nih.gov/25617572/), particularly regarding exer-gaming technologies such as Wii-Fit. With the emphasis placed on gamification of future IoT enabled weight management technologies, the omission of interactive gaming consoles, such as Wii-Fit should be strongly considered. 

Major comments:

1. One of the major conclusions from the study is that game-based technologies showed superiority, however, their analysis involves very few studies that evaluated gamification as a technique (Per Table IV). Based on the inclusion criteria noted in the paper, there are studies on devices aimed specifically exer-gaming devices (such as Wii-Fit) in adolescents that would be important to include in this analysis (see pub med link in paragraph above)

2. Table V is very informative but difficult to for the reader to get though. Is there another way to represent these data? Potentially consolidate?

Minor comments:

1. Citation needed at the end of the first Introduction paragraph

2. Introduction, second paragraph, end of second sentence: "apps" should be spelled out as applications (apps) and then shorthand may be used after.

3. Generally high use of acronyms and jargon. To improve readability, this should be clearly spelled out at first use each time.

4. Table I lists the titles of all 23 articles and is better suited for supplemental tables/Appendix rather than in the main text

5. Figures 1, 2, 3, 4, 5, and 6: Sort categories in graph by number (either ascending or descending; choose one and stick to that for remainder of bar graphs). Consider grouping categories with only 1 into one bar and specify "one each for the following: ____"

6. Figure 7 uses a new graph type to display the same data type. The circles are difficult to read. Consider the same chart/display for the same data type. 

7. Table VI does not add much to the aims of the manuscript—consider moving to supplemental information if included

 8. Paragraph right before "Strengths and limitations" header in discussion comments on the lack of adherence to CONSORT e-Health standards and importance of sticking to it. This is a good, and important, opportunity to spread knowledge. I.e. What is CONSORT e-Health and when should it be used.

Reviewer #3: The study addressed a very important issue, to manage children obesity through IoT but the reviewer opinion is that the manuscript cannot be published as it was submitted. It is not clear why most of the papers were discarded, what inclusion criterium they didn’t meet to be discarded. Tables are confused, long, difficult to read. It is not clear how they got the conclusion that gameable IoT are the most effective but decline of engagement was observed. It would be important to include in the conclusion what the goals of the study were, only change of weight? Change of lifestyle? Change of behavioral aspects? It is important to mention in the conclusion and perspectives if the studies were designed to increase or evaluate adherence and long-term effects. Is there any other way to present the information in more readable tables?

6. PLOS authors have the option to publish the peer review history of their article (what does this mean?). If published, this will include your full peer review and any attached files.

**Do you want your identity to be public for this peer review?** For information about this choice, including consent withdrawal, please see our Privacy Policy.

Reviewer #1: Yes: Ashwin Shreekant Sawant

Reviewer #2: No

Reviewer #3: Yes: Cleva Villanueva

---

## [Editor Report · Decision Letter 1]

28 Feb 2022

Internet of things-enabled technologies as an intervention for childhood obesity: A Systematic Review

PDIG-D-21-00086R1

Dear Mr Meinert,

We are pleased to inform you that your manuscript 'Internet of things-enabled technologies as an intervention for childhood obesity: A Systematic Review' has been provisionally accepted for publication in PLOS Digital Health.

Best regards,

Shelagh Mulvaney

Academic Editor

PLOS Digital Health

The authors have responded in full and appropriately to all reviewer comments and suggestions. The manuscript is more clear, the figures and tables are better organized, and the gamification point was clarified.